# Quantifying feral pig interactions to inform disease transmission networks

Tatiana Proboste[1]*, Abigail Turnlund[2], Andrew Bengsen[3], Matthew Gentle[4,5], Cameron Wilson[4], Lana Harriott[4], Richard A Fuller[6], Darren Marshall[7], Ricardo J Soares-Magalhaes[1]

[1]School of Veterinary Science, The University of Queensland, Brisbane, Australia; [2]The University of Queensland, School of Chemistry and Molecular Biosciences, Australian Centre for Ecogenomics, Brisbane, Australia; [3]NSW Department of Primary Industries, Vertebrate Pest Research Unit, Orange, Australia; [4]Pest Animal Research Centre, Biosecurity Queensland, Department of Agriculture and Fisheries, Toowoomba, Australia; [5]School of Sciences, University of Southern Queensland, Toowoomba, Australia; [6]School of the Environment, The University of Queensland, Brisbane, Australia; [7]Centre for Invasive Species Solutions, S. Bruce Australian Capital Territory, Bruce, Australia

## eLife Assessment

The authors aimed to quantify feral pig interactions in eastern Australia to inform disease transmission networks. They used GPS tracking data from 146 feral pigs across multiple locations to construct proximity-based social networks and analyze contact rates within and between pig social units. This **fundamental** study shows that targeting adult males in feral pig control programs could help global efforts to contain disease. The methods are **compelling** and the paper should be of interest to the fields of veterinary medicine, public health, and epidemiology.

*For correspondence: tatianaproboste@gmail.com

Competing interest: The authors declare that no competing interests exist.

**Abstract** Feral pigs threaten biodiversity in 54 countries and cause an estimated $120 billion in damages annually in the USA. They endanger over 600 native species and have driven 14 to extinction. Additionally, they pose a significant zoonotic disease risk, carrying pathogens such as Brucella, leptospirosis, and Japanese encephalitis. Understanding and controlling disease spread relies on models of social dynamics, but these vary widely across regions, limiting the transferability of findings from the USA and Europe to other locations like Australia. This study addresses this gap by analysing the social interactions of 146 GPS-tracked feral pigs in Australia using a proximity-based social network approach. Findings reveal that females exhibit stronger group cohesion, while males act as key connectors between groups. Contact rates are high within groups, facilitating rapid intra-group disease spread, whereas inter-group transmission is slower. Seasonal variations further impact dynamics, with increased contact in summer. These insights suggest that targeting adult males in control programs could help limit disease outbreaks. Given the rising economic and public health concerns associated with animal diseases, the study highlights the need for localized strategies based on feral pig social behaviour to enhance global control efforts.

## Introduction

Feral pigs (*Sus scrofa*) threaten biodiversity in 54 countries worldwide (*Risch et al., 2021*), and cause an estimated $120 billion of damage annually in the US (*Pimental, 2007*). Feral pigs imperil over 600

**eLife digest** Sometimes pigs and other domestic animals escape from farms and live independently in a similar way to their wild ancestors. Such animals and their offspring are referred to as "feral". In Australia, USA and many other countries across the world, feral pigs damage ecosystems and farmland, and carry Japanese encephalitis and other diseases that can spread to humans, livestock, and wildlife.

To understand and control these threats, researchers study how feral pigs move around and interact with each other, which is referred to as "social dynamics". However, previous studies have focused on groups of feral pigs living in Europe and may not be applicable to pigs living in Australia and other parts of the world. This lack of local data on social dynamics has made it difficult to optimise models of how diseases spread amongst feral pigs in certain areas. Optimising such models is crucial for enabling government agencies to prepare and respond to potential disease outbreaks in humans and other animals.

Proboste et al. analysed tracking data collected over a 7-year period from satellite transmitters attached to 146 feral pigs in eastern Australia. The analysis assessed how often the animals came close to each other and how this might influence the spread of diseases, revealing that females tended to remain within the same groups of pigs, while males were more likely to move between groups. The levels of contact between individual pigs varied across the year, with more contact during the summer months.

These findings show that male pigs play a key role in connecting different groups of feral pigs and may therefore be important targets for managing diseases in these animals. The findings also highlight that contact varies across the year, which might help pinpoint the best time to carry out management practices. In the future, this will help government agencies develop more effective strategies for managing feral pigs in Australia and other areas where their presence poses a risk to humans, livestock and native wildlife.

native species, and have directly driven 14 species to extinction (*Risch et al., 2021*), and cause significant economic loss to agricultural industries through negative impacts on water quality, extensive ground rooting, lamb predation, and crop and pasture consumption (*Bengsen et al., 2017*).

Feral pigs can be reservoir hosts of multiple infectious diseases of economic and/or animal health significance. This is particularly important for the continent of Australia, where many diseases are exotic, including foot-and-mouth disease (FMD), swine vesicular disease, Aujeszky's disease, and African and Classical swine fever (ASF and CSF). The potential economic impact of these diseases is substantial. For instance, an incursion of FMD is estimated to cost Australia $50 billion (*Buetre, 2013*) and the incursion of ASF can cost up to 2.5 billion (*Slatyer, 2023*). Feral pig populations in Australia can also be a source of zoonotic pathogens of public health importance including *Brucella suis* (*Pearson, 2012*), *Leptospira interrogans* (*Orr et al., 2022*), *Coxiella burnetii* (*Cooper et al., 2013*), and Japanese encephalitis virus (*Northern Territory Government, 2022*). Despite ongoing control measures that aim to reduce feral pig impacts, the threat of spillover of infectious diseases to domestic animals, wildlife, and human populations in Australia remains significant due to persistent feral pig populations.

## What is Australia doing to avoid the impact?

National preparedness for the incursion of diseases that are spread by feral pig activity relies on the development and validation of disease transmission models. Australia invests significant resources as part of the planning and prevention of new disease incursions and has developed the Australian Animal Disease Spread (AADIS) model (*Bradhurst et al., 2015*) for ASF and FMD. However, the performance of AADIS relies on adequate parameterisation of real-world feral pig contact rates, which is particularly challenging to obtain an estimate in wild pig populations (*Craft, 2015*). Currently, contact rate estimates needed to optimise the AADIS were obtained from European feral pig population studies (*Taylor et al., 2019*; *Taylor et al., 2021*; *Podgórski et al., 2018*), and remain unavailable for Australian feral pig populations. Locally relevant data on feral pigs contact rates will provide more reliable parameters to model the disease transmission rate in feral pig populations.

## Contact rates are imperative for disease models

To estimate disease spread, the force of infection is needed (β), which can be estimated by knowing the population of the contact rate (γ), the probability that contact is made with infected individuals, and the probability of pathogen transmission given a contact (*K*) (*Craft, 2015*; *Anderson and May, 1992*). Using individual movement network data, we can estimate the contact between individuals in wild populations. Contact rates between individual feral pigs are affected by feral pig social structure and spatial distribution (*Lloyd-Smith et al., 2005*). Historically, estimating contact rates from GPS tracking data of individuals has been challenging due to the need to set arbitrary time or distance thresholds to define a 'contact' or from limitations in the interval of the GPS collar recording location. New approaches have been recently developed to refine these calculations and to account for spatial autocorrelation, like continuous-time movement modelling (CTMM), which downscales the recording interval and improves the detection of contacts between individuals (*Calabrese, 2016*).

Understanding how animals move in space and time is critical to estimating the rate of infectious disease transmission and to optimising disease transmission models (*Albery et al., 2021*), to improve the confidence in projected impacts of modelled population control measures. Currently, there is a lack of understanding of contact rates between and within feral pigs, information that is necessary for the development of exotic disease model and contingency plans (*Spencer et al., 2005*). Therefore, it is vital to examine feral pig movement behaviours in a heterogeneous landscape, contact rates between individuals, differences in contact rates within and between pig social units (sounders), and to identify sex and age classes which are the most likely to transmit infections in case of an outbreak.

Contact heterogeneities for feral swine or wild boar have been estimated in different regions. In Poland, significant clustering of individuals was detected and differences in the duration of these associations depended on the sex of the individual (*Podgórski et al., 2014*). This reaffirms that dominant boars are primarily solitary, while reproductively active sows and their offspring largely remain in sounders (*Titus et al., 2022*; *Mayer, 2009*). Feral pig movement and interactions have also been explored with network analysis to estimate contact rates of populations in the USA (*Pepin et al., 2016*), Germany, and Italy (*Podórski et al., 2018*). For example, the US study found the contact rate between sounders was affected by the distance between the home range, with less contact when home ranges were separated by more than 2 km (*Pepin et al., 2016*). The European study found similar results and described that the most frequent association occurred at distances of 0–1 km and more sporadic association at more than 4 km (*Podórski et al., 2018*). However, such data have not been analysed for feral pigs in Australia.

This study aims to fill a continental gap in our understanding of feral pig social dynamics within and between sounders in eastern Australia, and to investigate differences between direct and indirect contacts. Our goal is to identify specific characteristics, such as demographic variables, which are common among individuals who are central to these networks. The findings from this research will enhance existing ASF models, as well as other pig disease transmission models, by providing empirical data on the structure of feral pig contact networks. Importantly, this approach allows us to identify key individuals that are central to these networks and, therefore, more likely to facilitate the spread of pathogens. This crucial information will greatly enhance the precision and relevance of disease modelling within feral pig populations to inform persistence, spread, and the design and implementation of management actions.

## Results

A total of 139,940 location fixes were collated from 146 GPS-monitored feral pigs, 61 females, and 85 males, tracked from 2017–2023 across QLD and NSW (*Figure 1*). Arcadia (QLD) population provided the most fixes (403,313) and the largest sample size (n=31), while Nap Nap (NSW) provided the least fixes (18,637) and had the smallest sample size (n=2). We obtained the most fixes during spring with 27%, 25% fixes during winter, and 24% fixes during autumn, and 24% fixes during summer season from 2017–2023 across QLD and NSW.

We compared the global network measures derived from thresholds of 2, 5, and 350 metres (*Figure 3—figure supplements 1 and 2*). The results revealed differences between the 2 and 5 metre thresholds in terms of average local transitivity, with an increase in the number of clusters as we raised the thresholds. Regarding edge density, which provides insight into the level of interconnectivity within

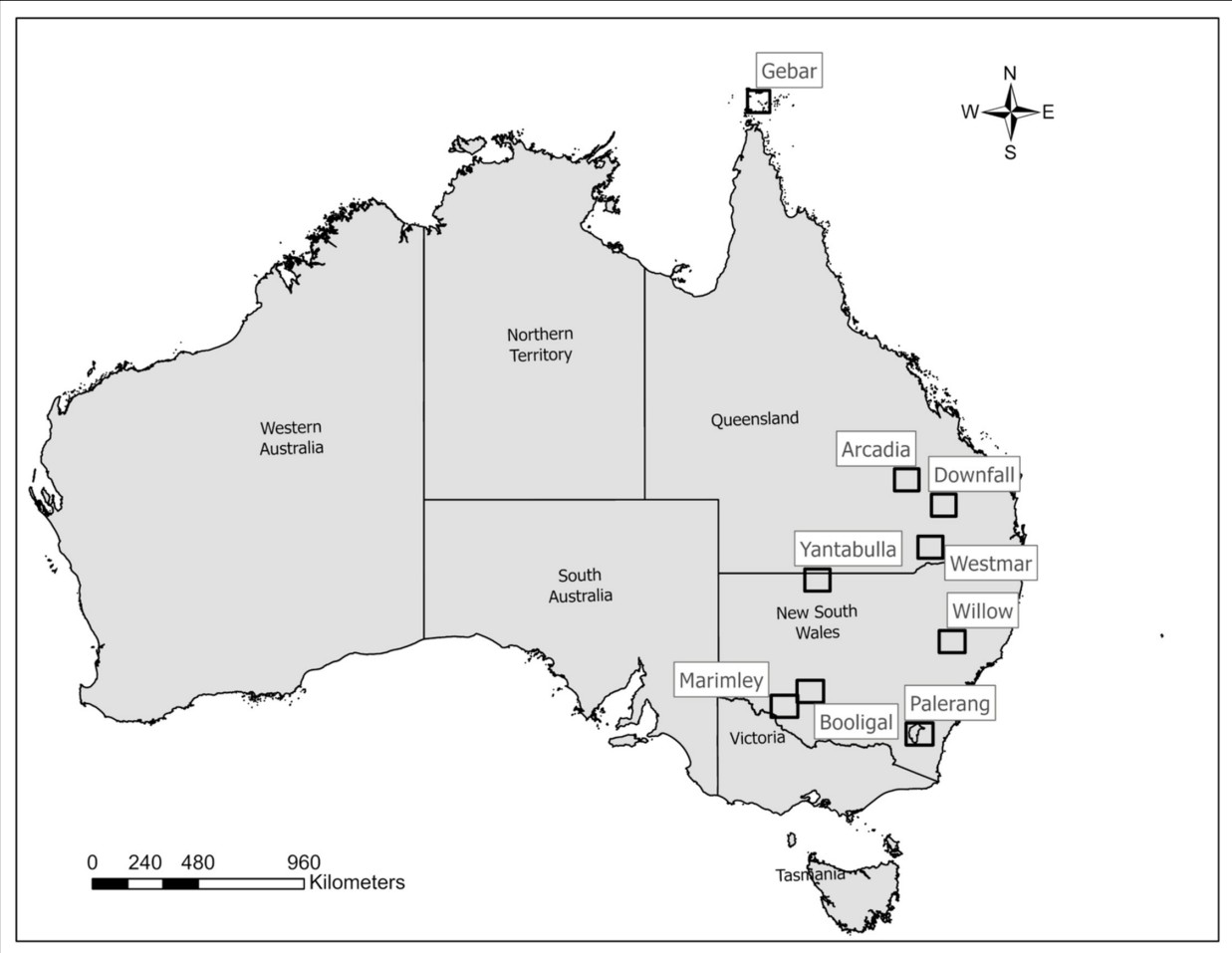

**Figure 1.** Map of Australia identifying the location (name of the population) of the study site.

the networks, our findings showed similar patterns across the thresholds, with a trend of increased interconnectivity as the threshold rose. The mean distance, which calculates the length of the shortest path for all possible pairs in the networks, showed minimal differences across the various thresholds.

The comparison of the global network measures between the direct and indirect networks (5 metres threshold), revealed that average local transitivity, edge density, and global transitivity were slightly higher for the indirect networks than the direct network (*Figure 3—figure supplement 1*).

Pigs within the Yantabulla population had a higher betweenness as well as a higher degree than pigs at other sites. Additionally, betweenness in males was, on average, 1.62 times greater than in females (*Figure 2*).

Comparison of node-level network measures between females and males for the direct network and indirect network based on a 5 metres threshold (*Figure 3*; *Figure 3—figure supplement 1*, *Figure 3—figure supplement 2* and *Figure 3—figure supplement 3*) revealed that males were positively associated with betweenness (log) (statistical significant for indirect network), while females were positively associated with higher strength (statistical significant for direct network) (*Table 1*).

## Home range

Home range overlap within dyads was greatest during summer, followed by autumn; however, this difference was not statistically significant (*Figure 4*).

## Contact rates

Most of the records of direct (96%) and indirect contact (69%) occurred between animals from the same sounder. Most observations of indirect contact within and between the sounders occurred

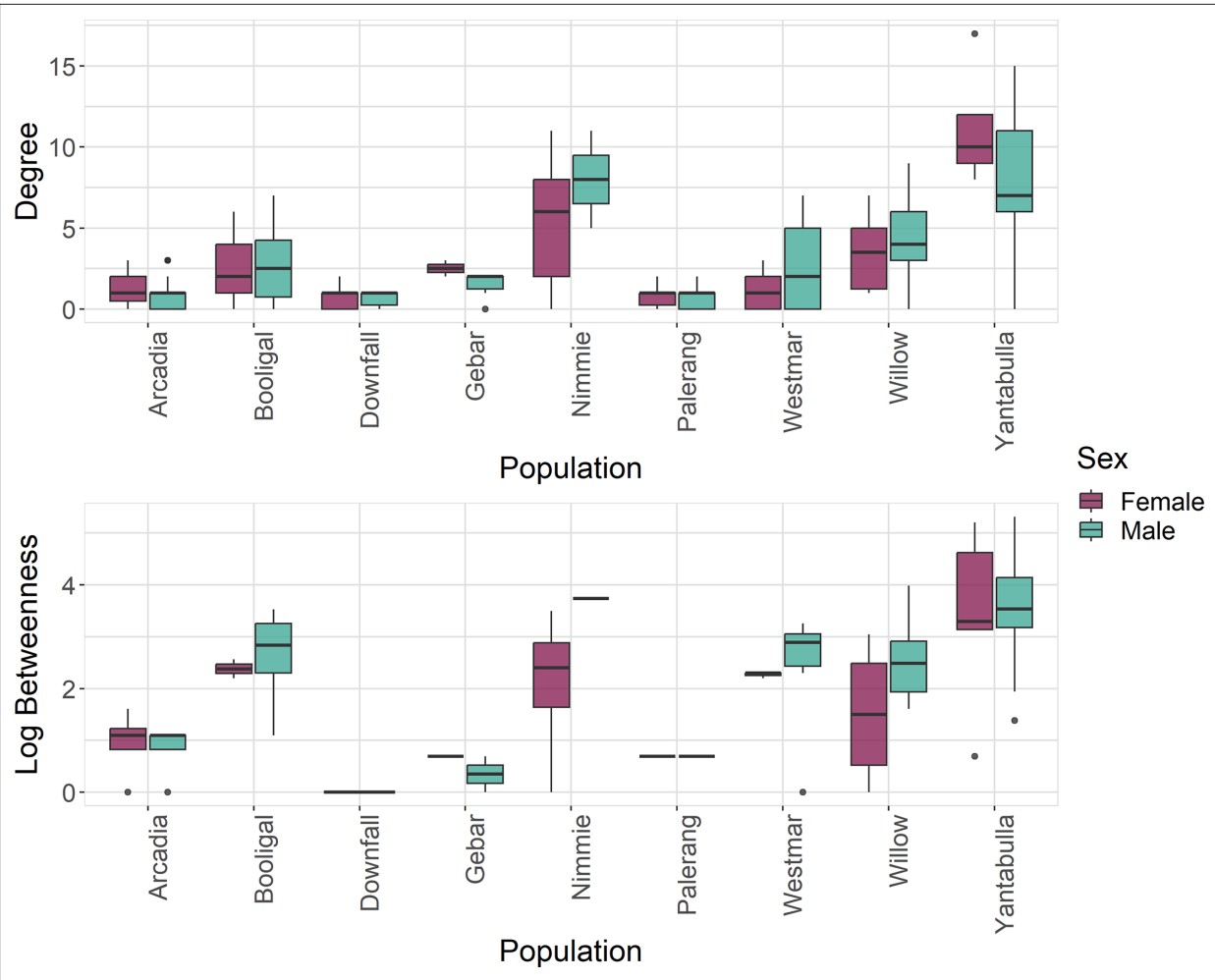

**Figure 2.** Betweenness (log) and degree measures at the individual level for each population and by sex, where green represents males and red represents females.

during winter and the least number of observations occurred during summer. We found a positive effect (p-values <0.001) on the contact rate within sounders compared to contact between sounders (overall contact rate in *Table 2* and *Figure 5*) and that the contact rate differed depending on the sex of the dyad.

Direct contact rates for female-female dyads across all interactions (within and between sounders) were 1.4 times greater than those for male-male dyads (estimate = −1.4; p<0.001) and 0.95 greater than those for male-female dyads (estimate = −0.95, p<0.001; *Figure 5a*).

Within sounders, we observed a similar trend, where female-female dyad were 1.56 times greater than male-male and 1.2 times greater than female-male, and contacts were more frequent during summer. We did not find a significant effect of sex of the dyad for contact rate between sounders, and contacts were more frequent during summer and winter compared to autumn.

For the indirect overall contacts (*Figure 5b*), we found that male-male and female-male contact were less frequent than female-female contact (p-values <0.001), and the contact rate was more frequent in summer compared to autumn (p-value <0.001). For the contact rate within sounders, we found a similar trend, with less frequent contact between female-male and male-male. Between sounders, male-male contact was less frequent compared to female-female contact, and summer had the highest contact frequency (p-value <0.001; *Table 2*). The mean direct and indirect rate per population, year, and season is detailed in *Supplementary file 1*, *Figure 5*. Distribution of the estimates and the effect of dyad sex and season in the contact rate (a) during direct contact and (b) for indirect contact. References are female-female for the sex of the dyad and autumn for the season.

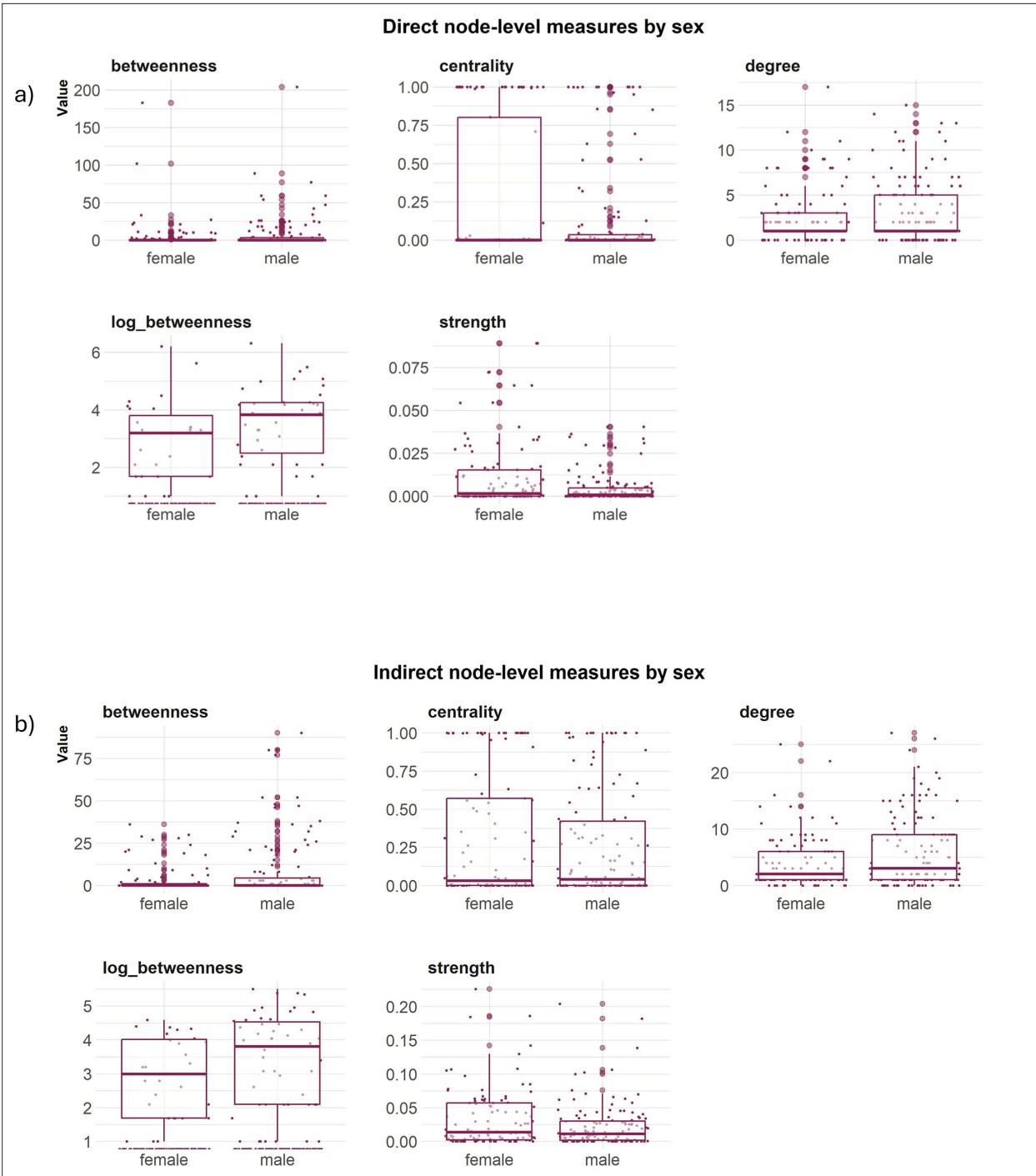

**Figure 3.** Node-level measures (5 m threshold), including betweenness, centrality, degree, log(betweenness) and strength by sex for (**a**) direct network and (**b**) indirect network.

The online version of this article includes the following figure supplement(s) for figure 3:

**Figure supplement 1.** Sensitivity analysis.

**Figure supplement 2.** Density comparison between different thresholds.

**Figure supplement 3.** Comparison between global network measures, including average local transitivity, edge density, global transitivity, mean distance and number of edges for direct and indirect networks using a 5 metres threshold.

**Table 1.** Differences between sex and network measures using Wilcoxon rank-sum test for direct and indirect contact.

| Network measures | Direct Contact Network | | | | Indirect Contact Networks | | | |
|---|---|---|---|---|---|---|---|---|
| | Mean rank (SD) | | W Statistic | p-value | Mean rank (SD) | | W Statistic | p-value |
| | Female | Male | | | Female | Male | | |
| Centrality | 0.263 (0.44) | 0.146 (0.32) | 7215.5 | 0.968 | 0.304 (0.41) | 0.256 (0.36) | 7204 | 0.985 |
| Betweenness | 4.77 (20.5) | 7.75 (23.0) | 6707.5 | 0.25 | 3.06 (7.59) | 8.51 (18.1) | 6162.5 | **0.021** |
| Strength | 0.012 (0.019) | 0.004 (0.009) | 8311.5 | **0.037** | 0.033 (0.043) | 0.023 (0.033) | 8086.5 | 0.097 |
| Degree | 2.56 (3.10) | 3.10 (3.60) | 6892 | 0.569 | 4.01 (4.55) | 5.93 (6.34) | 6165.5 | **0.054** |

SD: Standard deviation

Mean contact between sounders was generally lower than the mean contact within sounders. Within sounders, the highest mean contact occurred between dyads of females, and it was also higher for Arcadia (QLD) and Booligal (NSW) populations; both cases population the contact rate increased with the years. No contact was detected within sounders for two populations, Marimley and Nap Nap, as these populations had very small sample sizes (*Figure 6*, *Figure 6—figure supplement 1*) .

## Discussion

Models of disease spread provide important insights for preparedness and response. The incursion of an exotic enzootic disease such as African Swine Fever (ASF) or Foot-and-mouth Disease (FMD)

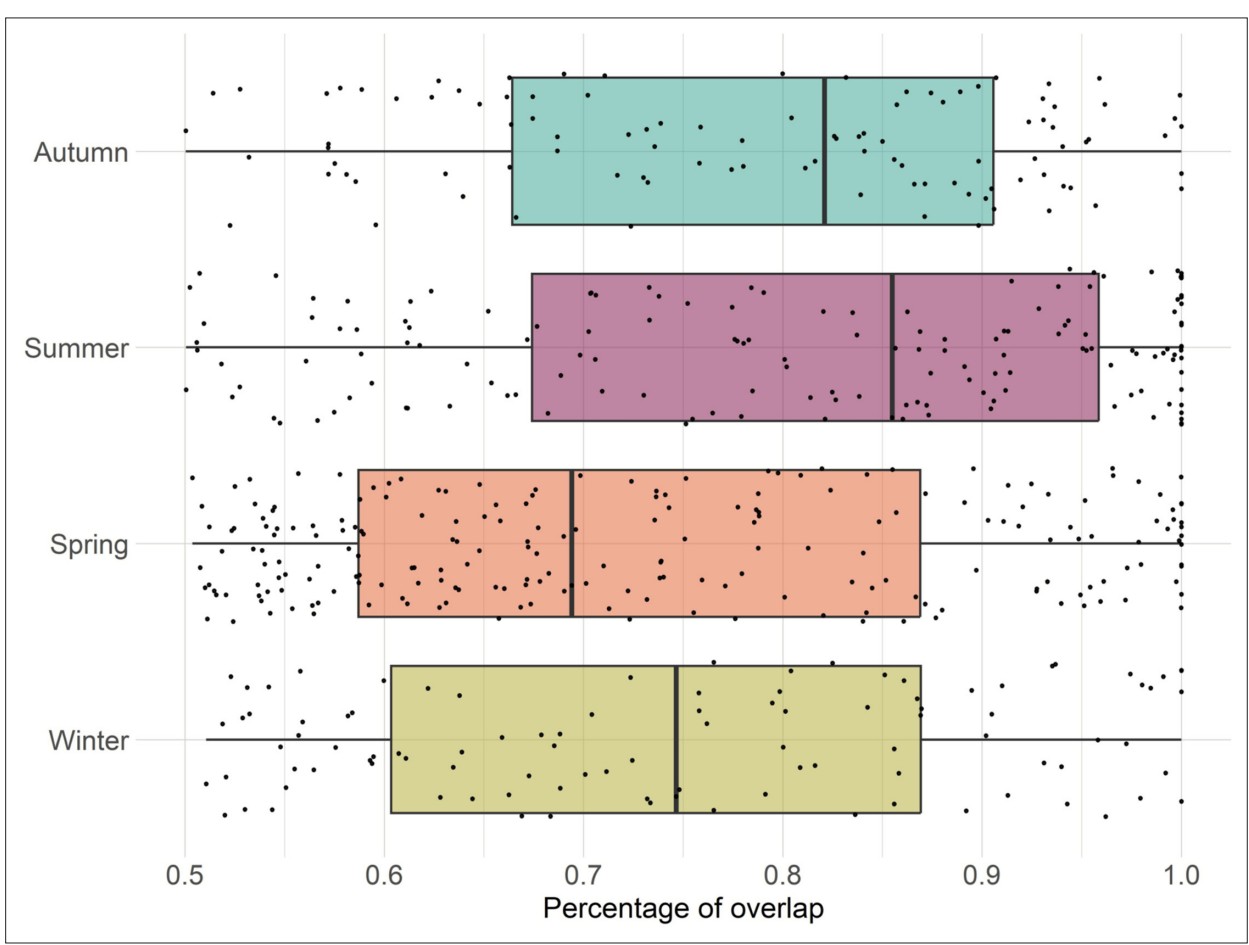

**Figure 4.** Home range overlaps per dyad for each season.

**Table 2.** Association between the mean contact rate within sounders, between sounders and overall and the distinct types of contacts (female-female, male-male, female-male) and seasons (autumn, spring, summer, winter), for direct and indirect contact calculated with 5 metres threshold.

| Response variable | Direct contact (within 5 min) | | | Indirect contact (within 5 d) | | |
|---|---|---|---|---|---|---|
| | Estimate | Std error | p-value | Estimate | Std error | p-value |
| Overall contact rates (within and between) | | | | | | |
| | 0.86 | 0.20 | <0.001 | 0.67 | 0.05 | <0.001 |
| | −1.40 | 0.33 | <0.001 | −0.45 | 0.09 | <0.001 |
| Contact type (within) | −0.95 | 0.27 | <0.001 | −0.31 | 0.07 | <0.001 |
| Male-male Female-male | 0.56 | 0.31 | 0.07 | 0.04 | 0.09 | 0.06 |
| Spring Summer | 0.87 | 0.30 | 0.004 | 0.31 | 0.08 | <0.001 |
| Winter | 0.46 | 0.30 | 0.13 | 0.03 | 0.09 | 0.77 |
| Contact rates within the sounder | | | | | | |
| | −1.59 | 0.45 | <0.001 | −0.36 | 0.12 | 0.001 |
| | −1.26 | 0.41 | <0.001 | −0.37 | 0.10 | <0.001 |
| Male-male Female-male | 0.40 | 0.32 | 0.28 | −0.15 | 0.11 | 0.18 |
| Spring Summer | 0.89 | 0.37 | 0.01 | −0.01 | 0.10 | 0.85 |
| Winter | 0.28 | 0.36 | 0.43 | −0.18 | 0.11 | 0.11 |
| Contact rates between sounders | | | | | | |
| | −0.17 | 0.51 | 0.73 | −0.44 | 0.13 | <0.001 |
| | 0.41 | 0.45 | 0.37 | −0.23 | 0.11 | 0.04 |
| Male-male Female-male | 1.18 | 0.46 | 0.13 | 0.23 | 0.13 | 0.08 |
| Spring Summer | 1.05 | 0.49 | 0.04 | 0.75 | 0.13 | <0.001 |
| Winter | 1.13 | 0.53 | 0.04 | 0.27 | 0.13 | 0.05 |

Std error: standard error

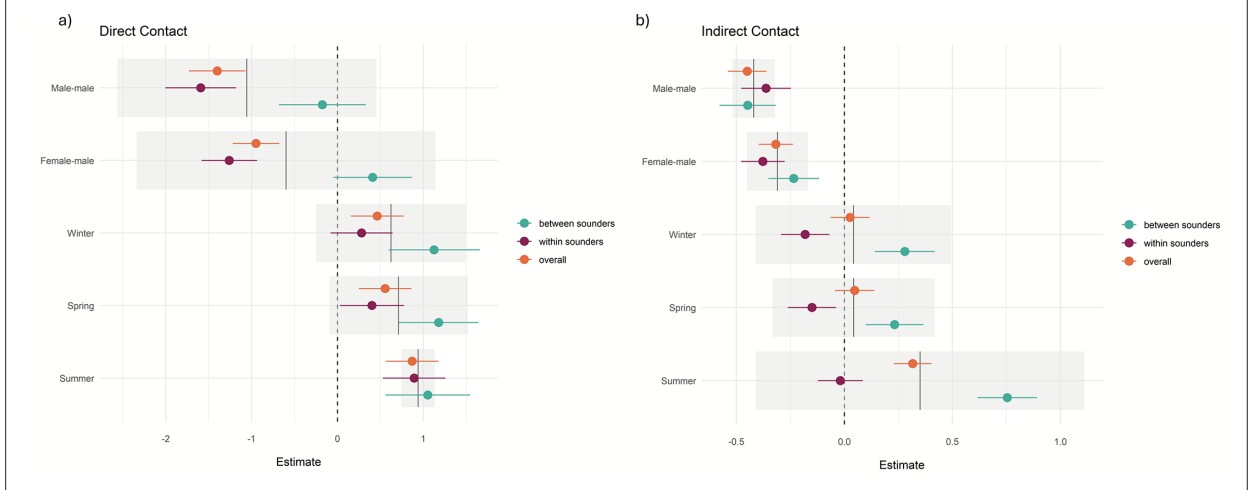

**Figure 5.** Diagram of direct and indirect means contact rate for each season for dyads between sounders and within sounders per year and population.

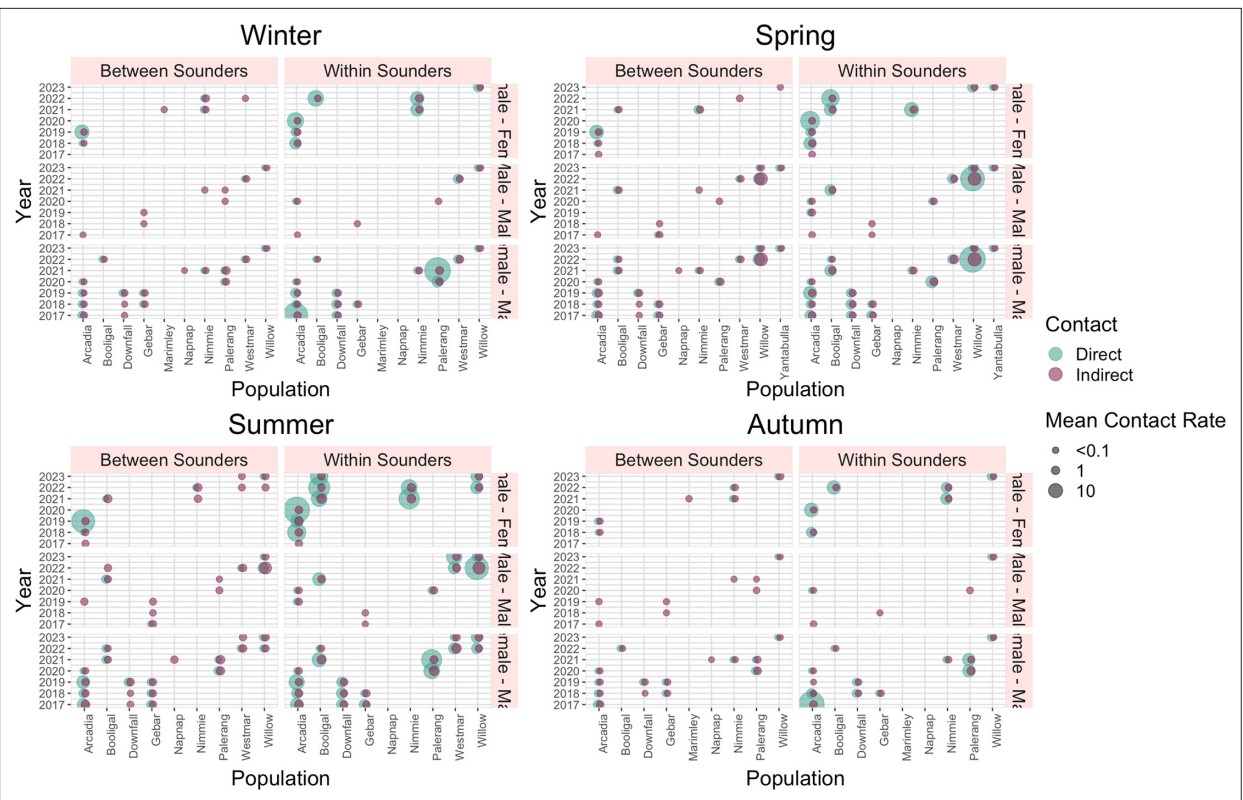

**Figure 6.** Diagram of direct and indirect mean contact rate for each season for dyads between sounders and within sounders per year and population.

The online version of this article includes the following figure supplement(s) for figure 6:

**Figure supplement 1.** Diagram of direct and indirect means contact rate for dyads between sounders and within sounders per year and population with a 2 metres threshold and a 5 metres threshold.

could result in devastating impacts on animal health and welfare, food production systems, and the economy in Australia. Feral pig populations are capable of spreading and maintaining these diseases in the environment, so it is critical to understand what level of contact occurs within and between feral pig populations to better inform disease preparedness. Here, we developed a proximity-based social network for feral pigs in eastern Australia, which elucidates the contact rate among individual pigs, both within and between sounders. The data procured from this network are critical for refining disease models and enhancing preparedness strategies in Australia. This study further revealed that the contact rate among feral pigs is dependent on two primary factors: the sex of the dyad and the season. It was observed that contacts were more prevalent among female dyads. Additionally, a seasonal pattern was discerned in the contact rate, with summer registering the highest frequency of contact. This research is the first of its kind in Australia, offering new insights into the social behaviour of feral pig populations.

## Network

In our comparison of global network measures between direct and indirect networks, we observed that the global transitivity, average local transitivity, and edge density were marginally higher for the indirect networks. This outcome aligns with our expectations, given that indirect networks permit a greater number of connections. This is due to the inclusion of associations between animals that occur when they share the same space within a window of 5 d. Understanding this indirect network is crucial for assessing the risk of pathogen dispersal, particularly when the pathogens exhibit prolonged persistence in the environment.

On the other hand, for the node-level networks measures, we found that females had higher strength in the direct network in comparison with males, likely due to greater group cohesion of females within a sounder (*Spencer et al., 2005*). In contrast, males had a positive effect on betweenness in

the indirect network, indicating that they are likely to connect independent groups through individuals with a higher tendency to move between different groups (*Farine and Whitehead, 2015*). The role of male feral pigs in connecting sounders should have important implications for the design of disease or population management strategies. For example, removal of adult females is typically seen as more important for sustained control programs aiming to reduce density-dependent damage caused by feral pigs because these individuals make the greatest contribution to population growth. However, the removal of adult males may be more important for control programs that aim to contain outbreaks of infectious disease because these individuals are likely to make the greatest contribution to disease transmission between sounders. Intensive population control can also cause surviving animals to change their spatial behaviour, resulting in increased risk of transmission or spillover (*Ham et al., 2019*).

## Home range

Previous research that studied four of the feral pig populations included in this study, found that home range was not affected by season (*Wilson et al., 2023a*). In this study, we did not explore the effect of season on feral pig home range, but rather focused on home range overlap between a dyad. The results of our analyses indicated that dyad (pair of individuals) had a greater home range overlap during summer and followed by autumn.

## Contact rate

The ability for disease to spread relies on many factors. Both direct and indirect contact with pathogens can have significant impact on disease transmission, and so it is important to consider both for epidemiological modelling. Direct contact, where an infected pig is in physical contact with a susceptible pig, is the primary method of transmission for diseases such as ASF and Food Mouth Disease. However, these diseases (e.g. ASF being viable for up to 5 d at 25 C° *Mazur-Panasiuk and Woźniakowski, 2020*), may also have indirect methods of transmission, for example, through ingestion of contaminated feed, survival of the pathogen in the soil or water sources or possible vectors, such as ticks. Understanding rates of indirect contacts are, therefore, highly relevant in understanding the risk of disease transmission.

Most **contact occurred within sounders**. We also found that the sex of the dyad statistically affects the contact rate. In our study, we found that female-female dyads were most frequent within the same sounder, which can be partly explained by the fact that sounders are composed mainly of adult females, sub-adults, and juveniles (*Spencer et al., 2005*; *Boitani et al., 1994*; *Gabor et al., 1999*). Our results differ from a previous study on feral pigs in the United States, which described no significant effect of sex for either direct contact or indirect contact (*Pepin et al., 2016*). This could be partly explained by the tendency of females to travel significantly less distance per day, with an average of 3.6 km compared to 4.9 km for males and, therefore, having more frequent female-female contacts (*Wilson et al., 2023b*).

The estimated contact rate was higher within animals from the same sounder than between sounder, which mean that disease could spread quickly within a sounder and may take longer between sounders. This result has been previously reported for wild boar populations in Europe, where they also found that contact rates between sounders were dependent on the distance between the groups (*Podgórski et al., 2018*). Moreover, within the sounder, the sex of the dyad played a significant role in the direct contact rate, while between sounder this dyad's sex is not statistically significant.

For **contact rates between sounders**, we found that sex influenced the rate for indirect contact only, with fewer contacts between males when compared to female-female. These results are in contrast to a study on wild boars in Europe that described no effect of animal sex on the association between sounders (*Podgórski et al., 2018*). For both direct and indirect contact rates between sounders, male-male dyad and female-male dyad had a negative effect.

We found that the contact rate within and between sounders was dynamic across the different seasons, which is similar to what has previously been described in wild boars (*Yang et al., 2021*). We found significant differences between seasons with a higher contact rate in summer. This difference in seasonality is critical for understanding management of feral pig populations, but are also important for optimising disease transmission models such as the AADIS model (*Bradhurst, 2021*). The incursion and transmission of a new virus into a susceptible feral pig population may spread differently under

varying conditions depending on season. For example, controlling an outbreak during summer would potentially require more resources than an outbreak in other seasons due to the higher number of contacts between individuals during summer. The findings from this study suggest that the incursion of a virus in summer would have a higher impact on the feral pig population (between and within sounders) than comparatively if it were introduced in autumn.

### Limitations

Different populations in our study had varying numbers of collared individuals, with some populations having only two individuals at certain times. This variability in sample size across populations is a limitation when interpreting the results. Small populations are often the result of a few individuals being trapped and collared, and this does not necessarily reflect the actual number of individuals in those groups. This issue creates bias when calculating connections between individuals. For those populations with only a few collared animals, we may observe very few connections and low network measures. However, this could be due to sparse information rather than a true reflection of the population's structure.

One of the populations (Gebar) is on a small island, which also may incorporate some bias to the network for this location as individuals in that population may have higher contact due to the limited space available.

### Conclusion

Our study highlights the role of sex and seasonality in contact rates, with implications for disease spread and control strategies. Specifically, the higher betweenness of males suggests their crucial role in disease transmission between groups, indicating that removal of adult males may be more effective in containing disease outbreaks. Furthermore, the higher contact rate observed in summer suggests that a virus introduced during this season could have a more significant impact on feral pig populations. These findings have important implications for disease modelling and management of feral pig populations in Australia, aiding in the development of more effective and targeted strategies for disease control and population management.

## Methods

### Ethical approval for the study

Data from feral pigs fitted with GPS-tracking collars were provided by the Queensland Department of Agriculture and Fisheries (DAF) and the New South Wales Department of Primary Industries (DPI). Feral pig GPS-tracking for animals in the state of New South Wales was approved by Animal Research Authority ORA 21-24-003 issued by the NSW Department of Primary Industries Orange Animal Ethics Committee. Animal use for research in Queensland were conducted under approval by the University of New England Animal Ethics Committee (AEC 16–115, AEC 20–023, and AEC 22–056).

### Data sources and data management

We used geolocation data of 74 individual feral pigs from NSW and 72 animals from QLD, for a total of 146 feral pigs collared between 2017 and 2023 for QLD and between 2021 and 2023 for NSW (*Figure 1*). These animals were from a total of 11 different populations from NSW (n=6) and QLD (n=5). The recording intervals for individuals collared in QLD was 30 min and data were cleaned by removing points recorded prior to capture, during the first 2 d after collaring, post mortality, and any inaccuracies (*Wilson et al., 2023a*). NSW individuals had 1 hr recording intervals and data were filtered the same as QLD individuals with the additional removal of points with incomplete or low-quality fixes, and locations confirmed by less than three satellites (*Bjørneraas et al., 2010*).

Contacts between individuals can be underestimated when the temporal resolution of GPS data is coarser than 30 min (*Yang et al., 2023*), therefore, we incorporated continuous-time movement models (CTMMs) for both NSW and QLD datasets to fit GPS tracking data to infer trajectories at 5 min intervals (*Calabrese, 2016*; *Yang et al., 2023*). This was done by fitting the GPS data using Ornstein-Uhlenbeck F (OUF) with ctmm.guess() function in the *ctmm* R package (*Calabrese, 2016*), which was then extrapolated in 5 min intervals using the predicted fit model (code adapted from *Yang et al., 2023*).

## Temporal and spatial thresholds for direct and indirect contact

Direct contact was defined when two individuals interacted either at 2, 5, or 350 metre buffers within a 5 min interval (*Yang et al., 2023*). A previous study used 350 metres as a spatial threshold (*Podgórski et al., 2018*), while other use the approximate average body length of an individual (*Yang et al., 2023*). ASF is estimated to remain infectious in the environment for 5 d (*Yang et al., 2023*) and, therefore, for the purpose of our study an indirect contact was defined when two individuals interacted within a 2 or 5 metre buffer within a 5 d interval. These spatiotemporal interaction thresholds aimed to represent indirect contact at populated locations (e.g. waterholes) that are potential hotspots for pathogen exposure. Temporal groups for direct and indirect contact were calculated with R package spatsoc (*Robitaille et al., 2019*) group_times() function using the temporal thresholds listed above (*Robitaille et al., 2019*). Then spatial thresholds were calculated with spatsoc group_pts() function using a threshold of 0.000045 degrees (~5 metres) that was split by pig population, year, and season (*Robitaille et al., 2019*). In the networks, each individual represents a node, and the contact between two individuals represents an edge. A spatial matrix was then created for direct and indirect contact by calculating the edge distances of each individual with the r package spatsoc edge_dist() function using the same degree threshold and split parameters as listed above (*Robitaille et al., 2019*).

## Network analysis

To estimate the contact between individuals, networks were created for each population and year (*Farine and Whitehead, 2015*; *Dougherty et al., 2018*). Proximity-based individual networks were built using the location of each individual in both space and time using the *spatsoc* R package (*Robitaille et al., 2019*). Spatial-temporal matrices calculated for specific indirect and direct contact thresholds, described above, were used to create separate networks for indirect and direct contact. First, individual matrixes were created from the spatial-temporal matrices with r package spatsoc get_gbi() function, and then a network was built with the get_network() function from the *asnipe* R package using SRI (simple ratio index) as the association index (*Farine, 2013*). An undirected weighted network adjacency matrix, which assigns nodes different values depending on the number of associations the connected nodes have, was created for each population in each recorded year with the R package (*Csardi and Nepusz, 2006*).

Networks were visualised with the R package *ggraph* (*Pedersen, 2022*). Node-level network metrics were further calculated with the R package *igraph* (*Csardi and Nepusz, 2006*) including degree centrality, strength, and betweenness for each node. Degree centrality quantifies the number of connections an individual has. The more connections an individual has, the higher their degree of centrality. Strength, on the other hand, encapsulates the robustness of these connections. The more time two individuals spend connected, the greater the strength (*Farine and Whitehead, 2015*). Lastly, node betweenness measures the number of paths that pass through a node (or individual) to connect two other nodes. In other words, it quantifies how often an individual serves as the quickest link between two other individuals in a network. Betweenness is a crucial measure in the context of disease transmission, as it indicates how frequently an individual could potentially transmit a disease between two others. Global network measures were also calculated, including, average local transitivity, global transitivity, edge density, number of nodes, mean distance, and number of edges for each population and by year. Transitivity measures the likelihood that the adjacent edges of a node are connected (ratio of count of triangles). The local transitivity gives a measure of how interconnected a node's neighbours are, and the global transitivity measure the overall level of transitivity in the network. (*Csardi and Nepusz, 2006*). We also calculated these measures for the three thresholds (2, 5 and 350m) and compared the differences between the structures of the networks based on the thresholds.

## Home range estimation

Home range is defined as the area utilised by each animal for its normal activities of foraging, mating, and caring for young (*Burt, 1943*). Average seasonal home ranges for each individual in each recorded season (Summer: December-February; Autumn: March-May; Winter: June-August; Spring: September-November) were estimated for populations with more than three individuals. Home ranges were determined by first calculating the kernel density with 95% confidence with the *amt* R package hr_kde() function (*Signer et al., 2019*). The home range overlap was then determined by comparing

the proportions of kernel densities that overlap between each pair of individuals using the amt R package hr_overlap() function (*Signer et al., 2019*). To determine if dyads were comparing individuals within or between sounders, dyads with home range overlaps over 50% were considered to belong within the same sounder, and dyads with home range overlaps under 50% were considered to belong to different sounders (*Gabor et al., 1999*).

## Contact rates

Direct and indirect contact rates within and between sounders at each site, as well as overall expected contact rate, were estimated for each year, season, and pig population. The contact rate refers to the frequency at which two specific individuals come into contact per unit of time. Contact rates were determined by first calculating how many days each individual was recorded in the direct and indirect spatial and temporal threshold and then determining if the dyad being compared was within or between different sounders, and the type of contact based on sex (female-female, male-male, or female-male). Contact rates were then calculated for each dyad by dividing the number of contact observations from the dyad distance matrix by the minimum number of days for the individual with the least recordings. Mean direct and indirect contact rates and standard deviation were calculated for each type of contact or dyad, which refers to a pair of individuals (female-female, male-male, female-male) between and within sounders for each population for every season and year.

## Statistical analyses

We used three linear mixed-effects models to estimate the effects of sex, season, and type of contact (between or within sounders) on direct contact rates. Parameters were selected based on the results of null hypothesis testing. The first model specified overall contact rate (between and within sounders) as the dependent variable and included fixed effects for type of contact (between or within sounder), sex of the dyad, and season. Pig population was included as a random effect. The second model used direct contact rates within sounders as the dependent variable and included the same fixed and random effects as the first model, excluding type of contact. The third model included the same fixed and random effects as the second model, using direct contact rates between sounders as the dependent variable. We then repeated the process using indirect contact rates for the dependent variables. Models were fitted using the lme4 package (*Bates, 2015*) for R. To determine if sex influences the network measures (degree, centrality, betweenness, and strength) we performed a Wilcoxon ran-sum test, using the networks build with 5 metres threshold and for direct and indirect networks. The code can be found in GitHub.

## Acknowledgements

The authors want to thank Southern Queensland Landscapes for providing data pertaining to this study. We acknowledge the HEAL (Healthy Environments and Lives) National Research Network, which receives funding from the National Health and Medical Research Council (NHMRC) Special Initiative in Human Health and Environmental Change (2008937). We acknowledge NSW Local Land Services and Southern Queensland Landscapes for their help with the trapping and collaring of the animals.

## Additional information

### Funding

| Funder | Grant reference number | Author |
| --- | --- | --- |
| National Health and Medical Research Council | 2008937 | Ricardo J Soares-Magalhaes |
| NSW Department of Primary Industries | | Tatiana Proboste |

The funders had no role in study design, data collection and interpretation, or the decision to submit the work for publication.

## Author contributions

Tatiana Proboste, Conceptualization, Formal analysis, Investigation, Methodology, Writing – original draft, Project administration, Writing – review and editing; Abigail Turnlund, Data curation, Formal analysis, Writing – review and editing; Andrew Bengsen, Resources, Data curation, Funding acquisition, Writing – review and editing; Matthew Gentle, Cameron Wilson, Lana Harriott, Conceptualization, Resources, Data curation, Writing – review and editing; Richard A Fuller, Supervision, Writing – review and editing; Darren Marshall, Resources, Data curation, Writing – review and editing; Ricardo J Soares-Magalhaes, Conceptualization, Resources, Supervision, Writing – review and editing

## Author ORCIDs

Tatiana Proboste ⓘ https://orcid.org/0000-0002-5274-7179

## Ethics

All animal handling and data collection procedures were approved by the University of New England Animal Ethics Committee (AEC 16-115, AEC 20-023, AEC 22-056) and the NSW Department of Primary Industries Animal Ethics Committee (ORA 21-24-003), and were conducted in accordance with all relevant institutional and national guidelines for animal research.

Reviewer #2 (Public review): https://doi.org/10.7554/eLife.102643.3.sa1
Reviewer #3 (Public review): https://doi.org/10.7554/eLife.102643.3.sa2
Author response https://doi.org/10.7554/eLife.102643.3.sa3

# Additional files

## Supplementary files

Supplementary file 1. Summary of the mean direct contact rate per population, year, season, and type of dyad.

MDAR checklist

## Data availability

Animal location data are classified as sensitive due to biosecurity, and privacy concerns. Public sharing of these data could pose risks to animals, including potential habitat disturbance or unauthorized use. Additionally, the dataset is the intellectual property of the Department of Primary Industries and Regional Development (DPIRD) and the Department of Agriculture and Fisheries (DAF), and access is restricted under institutional and regulatory policies. The aggregated data and summary of contact between individuals by population, year, and season have been shared in Supplementary file 1. Specific locations of the populations made available upon request. Interested researchers should contact the corresponding author to discuss data access under appropriate agreements. All modelling code used in this study is openly available on GitHub: https://github.com/Tatianaproboste/Feral-Pig-Interactions (copy archived at *Proboste, 2025*). We also include code to generate a dataset to be able to run the code with other data. All figures were done in R using the codes stored in the GitHub repository.

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
