## [Editor Report · eLife Assessment]

The authors aimed to quantify feral pig interactions in eastern Australia to inform disease transmission networks. They used GPS tracking data from 146 feral pigs across multiple locations to construct proximity-based social networks and analyze contact rates within and between pig social units. This **fundamental** study shows that targeting adult males in feral pig control programs could help global efforts to contain disease. The methods are **compelling** and the paper should be of interest to the fields of veterinary medicine, public health, and epidemiology.

---

## [Referee Report · Reviewer #2 (Public review)]

Summary:

The paper attempts to elucidate how feral (wild) pigs cause distortion of the environment in over 54 countries of the world, particularly Australia.

The paper displays proof that over $120 billion worth of facilities were destroyed annually in the United States of America.

The authors have tried to infer that the findings of their work were fundamental and possessing a compelling strength of evidence.

Strengths:

(1) Clearly stating feral (wild) pigs as a problem in the environment.

(2) Stating how 54 countries were affected by the feral pigs.

(3) Mentioning how $120 billion was lost in the US, annually, as a result of the activities of the feral pigs.

(4) Amplifying the fact that 14 species of animals were being driven into extinction by the feral pigs.

(5) Feral pigs possessing zoonotic abilities.

(6) Feral pigs acting as reservoirs for endemic diseases like brucellosis and leptospirosis.

(7) Understanding disease patterns by the social dynamics of feral pig interactions.

(8) The use of 146 GPS-monitored feral pigs to establish their social interaction among themselves.

Weaknesses:

None, as the weaknesses had been already addressed.

---

## [Referee Report · Reviewer #3 (Public review)]

Summary:

The authors sought to understand social interactions both within and between groups of feral pigs, with the intent of applying their findings to models of disease transmission. The authors analyzed GPS tracking data from across various populations to determine patterns of contact that could support the transmission of a range of zoonotic and livestock diseases.

The analysis then focused on the effects of sex, group dynamics, and seasonal changes on contact rates that could be used to base targeted disease control strategies which would prioritize the removal of adult males for reducing intergroup disease transmission.

Strengths:

It utilized GPS tracking data from 146 feral pigs over several years, effectively capturing seasonal and spatial variation in the social behaviors of interest. Using proximity-based social network analysis, this work provides a highly resolved snapshot of contact rates and interactions both within and between groups, substantially improving research in wildlife disease transmission.

Results were highly useful and provided practical guidance for disease management, showing that control targeted at adult males could reduce intergroup disease transmission, hence providing an approach for the control of zoonotic and livestock diseases.

Weaknesses:

None, as the authors have already addressed the identified weaknesses.

---

## [Author Response]

The following is the authors’ response to the original reviews.

**Public Reviews:**

**Reviewer #1 (Public review):**
Summary:The authors aimed to quantify feral pig interactions in eastern Australia to inform disease transmission networks. They used GPS tracking data from 146 feral pigs across multiple locations to construct proximity-based social networks and analyse contact rates within and between pig social units.Strengths:(1) Addresses a critical knowledge gap in feral pig social dynamics in Australia.(2) Uses robust methodology combining GPS tracking and network analysis.(3) Provides valuable insights into sex-based and seasonal variations in contact rates.(4) Effectively contextualizes findings for disease transmission modeling and management.(5) Includes comprehensive ethical approval for animal research.(6) Utilizes data from multiple locations across eastern Australia, enhancing generalizability.Weaknesses:(1) Limited discussion of potential biases from varying sample sizes across populations

This is a really good comment, and we will address this in the discussion as one of the limitations of the study

(2) Some key figures are in supplementary materials rather than the main text.

We will move some of our supplementary material to the main text as suggested.

(3) Economic impact figures are from the US rather than Australia-specific data.

We included the impact figures that are available for Australia (for FDM), and we will include the estimated impact of ASF in Australia in the introduction.

(4) Rationale for spatial and temporal thresholds for defining contacts could be clearer.

We will improve the explanation of why we chose the spatial and temporal thresholds based on literature, the size of animals and GPS errors.

(5) Limited discussion of ethical considerations beyond basic animal ethics approval.

This research was conducted under an ethics committee's approval for collaring the feral pigs. This research is part of an ongoing pest management activity, and all the ethics approvals have been highlighted in the main manuscript.

The authors largely achieved their aims, with the results supporting their conclusions about the importance of sex and seasonality in feral pig contact networks. This work is likely to have a significant impact on feral pig management and disease control strategies in Australia, providing crucial data for refining disease transmission models.
**Reviewer #2 (Public review):**
Summary:The paper attempts to elucidate how feral (wild) pigs cause distortion of the environment in over 54 countries of the world, particularly Australia.The paper displays proof that over $120 billion worth of facilities were destroyed annually in the United States of America.The authors have tried to infer that the findings of their work were important and possess a convincing strength of evidence.Strengths:(1) Clearly stating feral (wild) pigs as a problem in the environment.(2) Stating how 54 countries were affected by the feral pigs.(3) Mentioning how $120 billion was lost in the US, annually, as a result of the activities of the feral pigs.(4) Amplifying the fact that 14 species of animals were being driven into extinction by the feral pigs.(5) Feral pigs possessing zoonotic abilities.(6) Feral pigs acting as reservoirs for endemic diseases like brucellosis and leptospirosis.(7) Understanding disease patterns by the social dynamics of feral pig interactions.(8) The use of 146 GPS-monitored feral pigs to establish their social interaction among themselves.Weaknesses:(1) Unclear explanation of the association of either the female or male feral pigs with each other, seasonally.

This will be better explained in the methods.

(2) The "abstract paragraph" was not justified.

We have justified the abstract paragraph as requested by the reviewer.

(3) Typographical errors in the abstract.

Typographical errors have been corrected in the Abstract.

**Reviewer #3 (Public review):**
Summary:The authors sought to understand social interactions both within and between groups of feral pigs, with the intent of applying their findings to models of disease transmission. The authors analyzed GPS tracking data from across various populations to determine patterns of contact that could support the transmission of a range of zoonotic and livestock diseases. The analysis then focused on the effects of sex, group dynamics, and seasonal changes on contact rates that could be used to base targeted disease control strategies that would prioritize the removal of adult males for reducing intergroup disease transmission.Strengths:It utilized GPS tracking data from 146 feral pigs over several years, effectively capturing seasonal and spatial variation in the social behaviors of interest. Using proximity-based social network analysis, this work provides a highly resolved snapshot of contact rates and interactions both within and between groups, substantially improving research in wildlife disease transmission. Results were highly useful and provided practical guidance for disease management, showing that control targeted at adult males could reduce intergroup disease transmission, hence providing an approach for the control of zoonotic and livestock diseases.Weaknesses:Despite their reliability, populations can be skewed by small sample sizes and limited generalizability due to specific environmental and demographic characteristics. Further validation is needed to account for additional environmental factors influencing social dynamics and contact rates.

This is a really good point, and we thank the reviewer for pointing out this issue. We will discuss the potential biases due to sample size in our discussion. We agree that environmental factors need to be incorporated and tested for their influence on social dynamics, and this will be added to the discussion as we have plans to expand this research and conduct, the analysis to determine if environmental factors are influencing social dynamics.

**Recommendations for the authors:**
Reviewer #1 (Recommendations for the authors):(1) Consider moving some key figures from supplementary materials to the main text to strengthen the presentation of results.

We included a new figure to strengthen the presentation of results (Figure 3a-b), which shows the node level measures by sex and for direct and indirect networks.

(2) Expand discussion of limitations, particularly addressing potential biases from varying sample sizes across populations.

We added more detail and clarity about this potential bias into the limitation section within the discussion: “Different populations in our study had varying numbers of collared individuals, with some populations having only two individuals at certain times. This variability in sample size across populations is a limitation when interpreting the results. Small populations are often the result of a few individuals being trapped and collared, and this does not necessarily reflect the actual number of individuals in those groups.” Moreover, while reviewing the effect of the potential bias, we found that a General Linear Mixed Effect Model (Table 1) was not optimal for analysing the effect of sex on the network measures, and therefore this analysis has been done again using a non-parametric test (Wilcoxon rank-sum test) for direct and indirect networks based on a 5 metres threshold (Table 1).

(3) If available, include Australia-specific economic impact data in the introduction.

We included the impact figures that are available for Australia (for FDM) in the introduction.

(4) Clarify the rationale for chosen spatial and temporal thresholds for defining contacts.

This has been added in the methodology: “Direct contact was defined when two individuals interacted either at 2, 5, or 350-metre buffers within a five-minute interval [36]. A previous study used 350 metres as a spatial threshold [16], while others use the approximate average body length of an individual [36]”

(5) Consider adding a brief discussion of ethical considerations beyond basic animal ethics approval, addressing aspects like animal welfare during collaring and potential environmental impacts.

Feral pigs are an invasive species in Australia, and managing their population is crucial to protecting native ecosystems. The trapping and collaring of these animals have been conducted following the stringent animal welfare requirements necessary to obtain animal ethics approval in Australia. However, it is important to consider the broader ethical implications. Animal welfare during collaring is a critical aspect and involves minimising stress and physical harm to the animals. The collars used are lightweight and properly fitted only on adults due to welfare issues collaring juveniles.

(6) Add a statement about data availability/accessibility.

The GPS data cannot be shared; however, the R codes will be deposited in GitHub (https://github.com/Tatianaproboste/Feral-Pig-Interactions) and the link has been added in the final version.

(7) Expand on the implications of seasonal variation in contact rates for disease management strategies in the discussion.

We have added this information in the discussion: “For example, controlling an outbreak during summer would potentially require more resources than an outbreak in other seasons due to the higher number of contact between individuals during summer.”

**Reviewer #2 (Recommendations for the authors):**
The typographical errors in the abstract to be corrected are:(1) Line 22: Remove the "are" before "threaten".

This has been corrected.

(2) Line 24: Replace the "to" before "extinction" with "into".

This has been corrected.

(3) Line 28: Rephrase the sentence.

‘Yet social dynamics are known to vary enormously from place to place, so knowledge generated for example in USA and Europe might not easily transfer to locations such as Australia.’

(3) Line 29: Insert a "comma" after "Here".

This has been corrected.

(4) Lines 33 -34: Explain, clearly, the contact rates; is it between females to females or females to males?

We have improved this phrase and now it reads: “…. with females demonstrating higher group cohesion (female-female) and males acting as crucial connectors between independent groups.”

(5) Line 36: Make yourselves clear about what you mean by "targeting adult male".

We believe “targeting adult males” is correct in this context.

**Reviewer #3 (Recommendations for the authors):**
(1) Line 22 and 44, I think are threaten "are" should be removed for better clarity.

This has been corrected.

(2) Line 71, the source and not "force" of infection.

The force of infection is correct here.

(3) Line 72, population "of".

This has been corrected.

(4) Under statistical analysis, the software version should be included.

R has changed to multiple versions since we started this analysis.

(5) Terminological consistency: as far as possible try to be consistent with the terms used in the text, such as using "contact rate" instead of "interaction rate" in order not to puzzle the readers.

We have changed most of the “interactions” to “contact” instead as suggested.

(6) Correct Typos: Identify typos and grammatical inconsistencies of any kind, especially in those complex sentences that may be hard to follow.

The typos have been checked.

(7) Under the methodology, briefly describe why specific thresholds were chosen and any limitations.

We added the following into the method: “Direct contact was defined when two individuals interacted either at 2, 5, or 350-metre buffers within a five-minute interval [36]. A previous study used 350 metres as a spatial threshold [16], while others use the approximate average body length of an individual [36]”

(8) The discussion should be strengthened by drawing clear links between the findings and actionable management strategies.

We have strengthened the discussion by adding more specific actionable management strategies. For example, controlling an outbreak during summer would potentially require more resources than an outbreak in other seasons due to the higher number of contacts between individuals during summer.

(9) Did you consider additional environmental factors, such as rainfall, food availability, or habitat features, to better understand how these influence seasonal variations in pig interactions and contact rates?

This is something that we have in mind and will explore in future research. This has been partially explored but is based on how environmental factors and seasons affect the home range (Wilson et al 2023).

(10) Figure Legends: Add more detailed descriptions in figure legends, especially for those figures showing network metrics or contact rates.

More information has been added to the figure legends.

(11) The paper includes too many figures, and thus, it is recommended to simplify or merge some figures where appropriate. In particular, this is recommended for those figures that plot more network measures across thresholds. Adding clear, summarized captions with interpretation on threshold and measure significance would be a great help in interpreting complicated visualizations.

The figure that shows the comparison between global network measures, including average local transitivity, edge density, global transitivity, mean distance and number of edges for direct and indirect networks has been moved to supplementary material (Figure S3). We also included direct and indirect model-level measures by sex as in Figure 3 and improved the captions of the figures presented in the main document.